# Inferring Potential Cancer Driving Synonymous Variants

**DOI:** 10.3390/genes13050778

**Published:** 2022-04-27

**Authors:** Zishuo Zeng, Yana Bromberg

**Affiliations:** 1Department of Biochemistry and Microbiology, Rutgers University, New Brunswick, NJ 08873, USA; 2Department of Genetics, Rutgers University, Piscataway, NJ 08854, USA

**Keywords:** synonymous variants, sSNV, cancer drivers, somatic variants, variant functional impact

## Abstract

Synonymous single nucleotide variants (sSNVs) are often considered functionally silent, but a few cases of cancer-causing sSNVs have been reported. From available databases, we collected four categories of sSNVs: germline, somatic in normal tissues, somatic in cancerous tissues, and putative cancer drivers. We found that screening sSNVs for recurrence among patients, conservation of the affected genomic position, and synVep prediction (synVep is a machine learning-based sSNV effect predictor) recovers cancer driver variants (termed *proposed drivers*) and previously unknown putative cancer genes. Of the 2.9 million somatic sSNVs found in the COSMIC database, we identified 2111 proposed cancer driver sSNVs. Of these, 326 sSNVs could be further tagged for possible RNA splicing effects, RNA structural changes, and affected RBP motifs. This list of proposed cancer driver sSNVs provides computational guidance in prioritizing the experimental evaluation of synonymous mutations found in cancers. Furthermore, our list of novel potential cancer genes, galvanized by synonymous mutations, may highlight yet unexplored cancer mechanisms.

## 1. Introduction

Despite many years of concerted research efforts, cancer remains a major public health challenge with 19.3 million new cases and 10 million deaths worldwide in 2020 alone [1]. On the molecular level, cancer is caused by genetic variation, whether inherited or acquired via chance mutation, infection, or environmental exposure to toxins or ionizing radiation [2,3,4]. These changes result in aberrant and uncontrolled cell growth–a cancer hallmark [5].

Genetic mutations found in cancerous tissues can be designated as drivers or passengers [6]. Driver mutations are selectively advantageous to cancer development and growth (carcinogenesis), whereas passenger mutations are “by-products” of the carcinogenesis process. It is estimated that each tumor contains four or five driver mutations [7], while the vast majority of the remaining variants are passengers [8]. Differentiating driver mutations from passenger mutations remains an unsolved problem in cancer biology [9]. Identification of drivers typically involves multiple steps: identifying variants recurrent in different cancer samples, predicting the functional impact of these variants, and inspecting the variants’ underlying pathways and interaction networks–all in addition to experimental validation [10]. 

Cancer drivers range in size and effect from SNVs and small InDels (insertion or deletion of a few nucleotides) to genome rearrangement and copy number variation [11]. According to the International Cancer Genome Consortium (ICGC) data portal (https://dcc.icgc.org/ (accessed on 2 March 2022)) [12], the vast majority (>91%) of mutations found in cancer tumor samples are SNVs. OncoVar (ONCOgenic driver VARiants, https://oncovar.org/ (accessed on 2 March 2022)) is a recently developed database containing 20,162 cancer driver (missense, stop-gain, and stop-loss) mutations spanning 814 genes and 33 cancer types [13]. Note that SNVs located in the protein coding region may have different consequences: mutation that change the corresponding protein sequence are known as missense (or non-synonymous—nsSNV) variants, while those that, due to codon degeneracy, do not affect the protein sequence are synonymous (sSNV). The role of sSNVs in cancer is often overlooked [14] as for OncoVar and other databases (e.g., ICGC data portal [12]). However, sSNVs can have a variety of functional impacts on biological functionality (e.g., transcription, splicing, cotranslational folding) [15] and thus may also be cancer drivers. Supek et al. estimated that synonymous variants account for 6–8% of all SNV driver mutations in oncogenes [16]. In fact, multiple sSNVs in various genes and cancer types have been recognized as drivers, e.g., variants in BCL2L12/melanoma [17], VHL/hemangioblastoma [18], and BAP1/clear-cell renal cell carcinoma [19]. 

Here, we evaluated the effects of sSNVs from four categories: germline mutations, somatic mutations found in normal tissues, cancer somatic mutations, and putative cancer driver mutations. Based on the comparisons of variant effect predictions in these four categories of sSNVs, we demonstrated the utility of synVep [20], a machine learning-based method for sSNV effect prediction, in prioritizing putative cancer drivers. We then identified a list of putative cancer driver sSNVs and filtered this list via functional analysis to select 72 sSNVs, which are highly likely drivers in multiple cancer types, such as skin, large intestine, and liver, and should be among the priority candidates for experimental evaluation.

## 2. Materials and Methods

***sSNV collection.*** We consider four categories of sSNVs: germline sSNVs (denoted as *germline*), somatic sSNVs in normal tissue (*somatic normal*), somatic sSNVs in cancerous tissues (*somatic cancer*), and putative cancer driver sSNVs (*putative drivers*). *Germline*, *somatic normal*, and *somatic cancer* variants are obtained from the gnomAD project [21], SomaMutDB [22], and COSMIC [23] databases, respectively. gnomAD (Genome Aggregation Database, https://gnomad.broadinstitute.org/ (accessed on 9 November 2021)) houses data from large-scale sequencing efforts, identifying genomic variants from 16,708 genomes and 125,748 exomes; for the purposed of this paper, we only considered gnomAD exomes data, curated as described in our previous work [20]. SomaMutDB [22] (https://vijglab.einsteinmed.org/SomaMutDB/ (accessed on 7 December 2021)) contains 2.42 million SNVs and 0.12 million INDELs (insertions or deletions) identified from 19 normal human tissue samples or cell line types (e.g., brain, blood, breast, heart, lung, liver, skin) of 374 individuals. The Catalogue of Somatic Mutations In Cancer (COSMIC) [23] houses a collection of somatic mutations found in cancerous tissues. The latest release of COSMIC (v95) includes 41 million confirmed somatic coding point mutations (SNVs—single nucleotide polymorphisms) from genome wide screenings of 1.4 million cancer tissue samples from 37 cancer primary sites. To be consistent with *somatic cancer* sSNVs, we only selected tissues, but not cell lines, from SomaMutDB to create the *somatic normal* set of sSNVs. To compile *somatic cancer* sSNVs, we downloaded the “CosmicGenomeScreensMutantExport.tsv.gz” file from COSMIC (GRCh37, https://cancer.sanger.ac.uk/cosmic/download (accessed on 9 November 2021)) and filtered the data to be “Confirmed somatic variant” and “Substitution–coding silent”. We mapped the genomic positions of COSMIC and SomaMutDB variants to all possible human transcript-based positions of sSNVs from the synVep database [20]. 

The *putative drivers* were sSNVs selected from the SynMICdb database (Synonymous Mutations in Cancer database, http://synmicdb.dkfz.de/rsynmicdb/ (accessed on 7 December 2021)) [24]. SynMICdb houses 659,194 somatic sSNVs from COSMIC annotating their multiple aspects: whether the variant is in a cancer gene; variant frequency among healthy populations and in tumor samples; conservation of the affected genomic position (PhastCons [25]); pathogenicity/deleteriousness of the variant predicted by FATHMM-MKL [26] and CADD [27]; and the associated mRNA structural change predicted by remuRNA [28]. SynMICdb also provides SynMICdb scores, which are a heuristic combination of these annotations and are informative of the functional impact of sSNVs found in cancer; we selected variants with SynMICdb scores in or above the 95th percentile of all scores. 

***Gene-set enrichment analysis.*** We conducted gene-set enrichment analysis (GSEA) for gene ontology (GO) terms [29,30] on *germline*, *somatic normal*, *somatic cancer*, and *putative driver* sSNVs using clusterProfile [31] R package. The GSEA was performed for the top 10% of the genes with the highest normalized sSNV rate, i.e., the number of sSNVs in a gene divided by the coding length of that gene. Note that, to reduce GO term redundancy, we used the GO terms semantic similarity analysis [32] of the top identified GO terms removing lower-ranked terms that were >0.5 similar to higher ranked ones. 

***Cancer-associated genes.*** We downloaded 576 Cancer Gene Census [33] tier 1 genes from the COSMIC database (https://cancer.sanger.ac.uk/census (accessed on 7 December 2021)); tier 2, according to COSMIC, lacks extensive evidence and is thus not included. We extracted 51 cancer pathways from KEGG [34] (https://www.genome.jp/pathway/hsa05200 (accessed on 7 December 2021)) and identified 2210 corresponding genes using the clusterProfile [31] R package. We also obtained 217 disease ontology (DO) cancer terms from the supplementary data of Wu et al. [35] and identified 2895 corresponding genes using clusterProfile [31] R package. We term this collection of genes *cancer-associated*. In addition, we obtained from the literature [36] a set of 54 known oncogenes and 71 tumor suppressor genes.

***Proposing a novel list of cancer driver sSNVs.*** We applied the following criteria to all *somatic cancer* variants to propose a novel list of potential cancer driver sSNVs (denoted as *proposed driver*): (1) synVep score > 0.81, i.e., the median of synVep predictions for *putative driver* set; (2) GERP++ score [37] (from http://mendel.stanford.edu/SidowLab/downloads/gerp/ (accessed on 7 December 2021)) > 2.31, i.e., the median of GERP++ scores for the *putative driver* set; (3) located in a *cancer-associated* gene as defined above; (4) recurrent among cancer patients; here, we adopted the Sharma et al.’s approach [24] to define recurrence, i.e., mutations occurring more than once among different patients.

***Functional impact prediction for annotation of proposed driver variants.*** We used the CADD online server (https://cadd.gs.washington.edu/score (accessed on 3 March 2022)) annotations for GRCh37-v1.6 to retrieve CADD-splice (CADD v1.6) and spliceAI predictions for the *proposed driver* variants. For CADD-splice predictions, we considered sSNVs scoring > 15 to be splicing-disruptive (recommended cutoff at https://cadd.gs.washington.edu/info (accessed on 3 March 2022)). For spliceAI, we considered an sSNV to be splicing-disruptive if one of the four predictions generated (acceptor gain, acceptor loss, donor gain, and donor loss) was greater than 0.5. 

We used the RNAsnp [38] package for the prediction of changes to sSNV-affected RNA structures. As per the instructions (https://rth.dk/resources/rnasnp/software.php (accessed on 3 March 2022)), mode 1 was used for transcripts less than 200 nucleotides long, and mode 2 otherwise. Other parameters were set as default.

We further predicted all putative RBP motifs in all human protein coding transcripts (extracted from Ensembl BioMart assembly GRCh37 [39], https://figshare.com/articles/dataset/transcript_sequences_zip/19407530 (accessed on 3 March 2022)) using the online interface of the FIMO (Find Individual Motif Occurrences) [40] method from the MEME (Multiple Em for Motif Elicitation) suite [41] (https://meme-suite.org/meme/tools/fimo (accessed on 3 March 2022); all parameters set as default).

The human RBP motifs file (“Ray2013_rbp_Homo_sapiens.dna_encoded.meme” [42]) was obtained from the MEME motif database (https://meme-suite.org/meme/doc/download.html (accessed on 3 March 2022)). We then examined whether these extracted motifs overlapped with our *proposed driver* sSNV locations. 

We also mapped sSNVs to potential transcription factor binding sites (TFBS) via the SNP2TFBS [43] web server (https://ccg.epfl.ch/snp2tfbs/snpselect.php (accessed on 3 March 2022)). 

***Statistical analysis.*** Kruskal–Wallis test [44] was used as a non-parametric alternative to ANOVA to test whether the mean ranks of multiple groups are the same; post hoc pairwise comparison was performed with Dunn test [45] using FSA package [46] (https://cran.r-project.org/web/packages/FSA/index.html (accessed on 10 March 2022)) in R [47] (https://www.r-project.org/ (accessed on 10 March 2018)). 

## 3. Results and Discussion

### 3.1. sSNV-Affected Molecular Functions Differ by Variant Class

We evaluated the per-gene sSNV burden, i.e., the number of sSNVs per gene normalized by the length of the corresponding coding region (Methods), for all genes of all cancer patients in the COSMIC database. We found that sSNV burden of oncogenes and tumor suppressor genes (from [36]) does not differ significantly. However, both oncogenes and tumor suppressors have lower sSNV burden than either the Cancer Gene Census (CGC) [33] cancer genes or non-cancer genes (Appendix A). This unexpected observation may be due to the necessity of maintaining the specific (high or low) levels of functionality of oncogenes and TSGs in cancer development, while no such limitations/selections pressures are imposed on other genes. 

We also evaluated gene mutability overall by evaluating occurrence of other genetic variants. The numbers of nsSNVs per gene highly correlated with numbers of sSNVs (Pearson correlation = 0.86). However, the per gene nsSNV/sSNV ratio was also somewhat indicative of oncogenes. That is, the top 100 genes with highest nsSNV/sSNV ratio had more cancer genes (18%; oncogenes, tumor suppressors, and CGC) compared to the 100 genes with the lowest nsSNV/sSNV ratio (2%). The per gene nsSNV/sSNV derived from COSMIC database can be found in Appendix A. 

We further collected 4,221,244 *germline*, 54,368 *somatic normal*, 2,894,289 *somatic cancer*, and 27,878 *putative driver* sSNVs (Methods). For each of the sSNV categories, we calculated the normalized sSNV burden (highest ranked genes in Appendix A). As expected, genes containing the putative cancer drivers were heavily enriched in cancer association (61 of 100 were cancer genes). Cancer-associated genes were also found among germline and somatic cancer variant-enriched genes (3 of 100 genes each), but not in the somatic normal set. Curiously, the gene overlap among the four categories was minimal, indicating that somatic sSNVs affect different genes than germline sSNVs, as well as that mutation and selection mechanisms in cancer and normal tissues are also different. 

To compare the sSNVs across the four categories, we performed gene-set enrichment analyses (GSEA) for gene ontology (GO) [29,30] terms of the genes most-enriched in *germline*, *somatic normal*, *somatic cancer*, and *putative driver* sSNVs. Nine of the top ten *putative driver*-gene GO terms were unique to this set of variants, i.e., they were not in the top ten GO terms of *germline*, *somatic normal*, or *somatic cancer* genes (Figure 1), indicating that the biological functions of the *putative driver*-enriched genes are different from those of genes enriched in the other three categories of sSNVs. To evaluate the consistency of this observation, we conducted additional analyses with varying number of input genes (top 10%, 20%, and 30% genes with highest sSNV density), as well as varying number of GO terms (top 10, 20, and 30). The GO terms’ overlaps between *putative driver* and other groups were consistently low, ranging from 0 (e.g., overlap with *germline*, top 30% genes, top 10 GO terms) to 0.1 (e.g., overlap with *somatic normal*, top 20% genes, top 20 GO terms). Curiously, we also note that *germline* GO terms were very different from *somatic* ones, while *somatic cancer* and *normal*-enriched terms were somewhat similar. 

### 3.2. SynVep Variant Effect Scores Are Higher for Putative Drivers

*Putative driver* sSNVs are usually not observed in the general population, as was reflected by gnomAD, (Figure 2A). Furthermore, they were often localized to more conserved regions than the other three categories of sSNVs (Figure 2B). While reassuring, we note that this observation is trivial as mutation population frequency and conservation (PhastCons [25]) are included in the calculation of SynMICdb scores, which define putative driver variants. 

SynMICdb scores also includes functional prediction by CADD [27] (deleteriousness) and FATHMM-MKL [26] (pathogenicity). We previously developed synVep (synonymous Variant effect predictor) [20]—a machine learning-based method for predicting the likelihood of a human sSNV having an effect on the function of the corresponding gene product. synVep training relies on observation of sSNVs in human population, with a fundamental assumption that the unobserved sSNVs are enriched in functional effects. We demonstrated earlier that synVep can identify experimentally validated sSNV effects, pathogenic sSNVs, and splicing-disruptive sSNVs [20]. Importantly, synVep does not use conservation as a feature, thus providing information orthogonal to that of other functional impact predictors and to the SynMICdb score as a whole. From the synVep functional effect perspective, the *germline*, *somatic normal*, and *somatic cancer* variants may have an effect or not, but cancer driver mutations must have an effect. Thus, *putative driver* sSNVs were expected to have higher synVep scores, indicating variants that are more likely to have an effect, than variants of the other three categories of sSNVs (Figure 2C).

We note that higher synVep scores of the *putative driver* sSNVs may in part be due to their absence from the general population (Figure 2A)—a feature of most of the effect variants in synVep’s training set. To evaluate the effect of this potential bias, we removed all sSNVs labeled as observed in gnomAD from the somatic categories of variants and re-evaluated the synVep scores for the remaining data. The synVep predictions of *putative driver* sSNVs were still substantially higher than those of both *somatic normal* and *somatic cancer* sSNVs (Appendix A). 

Curiously, the *somatic normal* sSNVs, on average, scored higher than the *somatic cancer* sSNVs. This finding is in line with the fact that positive selection rules the likelihood of somatic variants [48,49] propagating throughout cells that make up individual tissues and, in order to be selected for, the variants need to have a molecular effect. In contrast, the vast majority of somatic mutations in cancerous tissues are passengers [8], as opposed to very few driver mutations, and are thus selectively neutral [50] having no or weak effect.

### 3.3. Screening sSNVs to Recover Cancer-Underlying Genes

Cancer genes are defined as those that can harbor mutations conferring growth advantage of tumor cells [51]. CGC [33] collects cancer genes with extensive evidence, but the discovery of all cancer genes is not yet close to completion [48]. For example, CGC genes only account for 54% and 70% of genes in KEGG cancer pathways [34] and in cancer ontologies [35], respectively. It is thus possible that mutations in non-CGC genes may be indirectly involved in causing cancer, by, e.g., contributing to initiation or progression of cancer or by enhancing the effects of cancer drivers [8,52,53]. With the increase in large-scale tumor sequencing, more data for analysis has become available and may identify additional cancer genes. However, different cancer gene identification methods produce different results and often fail to recover the previously identified cancer genes [54]. In other words, mutations labeled as non-drivers due to their localization to non-CGC genes may be incorrectly labeled, i.e., false negative. 

Given that most somatic mutations are random, recurrent mutations (same mutation in different cancer patients) are unlikely to occur by chance and are thus likely carcinogenic [10]. Evolutionary conservation is informative for prioritizing cancer drivers [55]. Furthermore, synVep, as we demonstrated earlier [20], is precise in differentiating sSNV molecular effects. Importantly, as conservation is not one of synVep’s feature, these two sSNV features are orthogonal. Following these observations, we identified four groups of genes based on whether they harbor certain types of sSNVs (Methods): (1) genes with non-recurrent sSNVs only, i.e., genes harboring recurrent sSNVs are excluded; (2) genes with recurrent sSNVs; (3) genes with recurrent sSNVs that are located at conserved positions; and (4) genes with recurrent sSNVs that are located at conserved positions and are scored high by synVep. We found that incorporation of recurrence, conservation, and synVep prediction filters identified genes that are more likely to be involved in cancer (Figure 3). For example, our most rigorous filtering identified 40% (229) of the 576 CGC genes in addition to another set of 4819 genes that are possibly cancer associated. In fact, 26% (*n* = 1329) of our genes were present in CGC, KEGG cancer pathways, or in the DO cancer gene list.

We also found that narrowing the lists to genes with more sSNVs that pass the above filters identifies more likely cancer genes (Figure 4). Note that, since different filters result in different distributions of variant counts, we use “>x percentile” to represent the top-ranking genes. For example, if the recurrence filter identifies 10 genes with 1, 1, 2, 2, 3, 3, 3, 3, 8, and 9 sSNVs, respectively, then the “>70-percentile” of the counts would include 8 and 9 sSNVs. The observation that genes with more sSNVs passing the filter are more likely cancer-associated is especially true for genes with recurrent variants. However, known cancer genes tend to have more non-recurrent sSNVs as well (Figure 4). One possible explanation is that the normal activity of cancer genes may also be disrupted by an accumulation of variants within the functional domains, whether the variants are recurrent or not [56,57]. 

Our results show that a high number of variants per gene that pass the recurrence, variant position conservation, and synVep filters can much better identify potential cancer genes than sSNV recurrence alone. There are 417 genes (Appendix A) containing > 17 sSNVs (> 90-percentile) that pass all of these filters. Among these genes, 40% (*n* = 166) are known to be cancer-associated, according to CGC, KEGG cancer pathways, or cancer DO. We expect that many of the remaining 251 genes may also be cancer-associated, although their mechanisms are yet not understood. As an example, consider three with the most sSNVs: *PCDH15*, *CELF4*, and *MYBPC1*. 

*PCDH15* encodes protocadherins, a group of calcium-dependent cell–cell adhesion protein [58]. It has been noted in earlier work as a potential marker for NK (natural killer)/T cell lymphomas [59]. Mutations in *PCDH15* have been identified in a whole-genome sequencing study [60] and an exome sequencing study [61] of prostate cancer. Another whole-exome sequencing study revealed that *PCDH15* harbored mutations associated with metastasis in ocular adnexal sebaceous carcinoma [62]. Furthermore, a genome-wide association study (GWAS) identified multiple loci in *PCDH15* to be significantly associated with acute myeloid leukemia [63].*CELF4* is one of the CELF proteins (CUGBP, ELAV-like family of proteins), which are a type of RNA-binding protein (RBP) with various roles in RNA regulation [64]. An earlier study identified an intronic *CELF4* germline variant associated with colorectal cancer risk [65]. Multiple other analyses found that *CELF4* can be used to prognose colorectal cancer [66,67,68]. Additionally, methylation of *CELF4* was proposed as a detection method for endometrial cancer [69].*MYBPC1* encodes a member of myosin-binding protein C family with a role in muscle contraction [70]. Significant differential expression of *MYBPC1* has been observed in tongue cancer [71], breast cancer [72,73], and prostate cancer [74]. Additionally, *MYBPC1* expression level was found to positively correlate with NK cell content [73].

Concordance between our findings and literature evidence for the likely involvement of our top ranked genes in cancer highlights the utility of our prioritization strategy, suggesting which unknown cancer-associated genes remain to be explored.

### 3.4. Selecting Novel Potential Cancer Driver sSNVs

As described above, we assume that cancer driver sSNVs can be identified by three filters: recurrence among cancer patients, affected genome position conservation, and synVep prediction on the sSNV impact. To identify a list of potential cancer driving sSNVs, we performed the following filtering: starting from 2,894,289 *somatic cancer* sSNVs from the COSMIC database, we applied four filters (recurrent variant, GERP++ score > 2.31, synVep prediction > 0.81, cancer-associated genes). We thus obtained 2111 (genomic position-based variants; mapping to 5021 transcript-based) sSNV candidates (Appendix A). These were evaluated for functional impact mechanism from three perspectives: mRNA alternative slicing, mRNA structural changes, as well as localization to RNA-binding protein (RBP) binding motifs; functional impacts of 326 sSNVs (genomic position-based; 609 transcript-based) were thus identified. A brief flowchart describing these processes is shown in Appendix A. We describe more detailed results of the functional impact evaluations below: 

*Splicing changes*: After transcription of a gene, splicing removes intronic sequences from the pre-mRNA molecule and/or joins exonic sequences. A primary transcript can be spliced into multiple mature mRNAs (known as alternative splicing) corresponding to different protein isoforms with varying functionalities [75]. Mutations can disrupt the splicing regulatory elements, resulting in aberrant splicing [76]. sSNV-induced aberrant splicing is common in multiple diseases [77,78], including cancers [16], and has been observed in many cancer genes, such as BRCA1 [79], BRCA2 [80], APC [81], and BAP1 [19]. CADD-splice [82] and spliceAI [83] are two state-of-the-art tools to predict splicing disruption induced by mutations. Of the 2111 *proposed driver* sSNVs, 136 (genomic coordinate-based; mapping to 222 transcript-based) sSNVs were predicted to be splicing-disruptive by CADD-splice or spliceAI (Appendix A) to be associated with aberrant splicing. In our set of variants, these putatively splicing-disrupting sSNVs affect multiple cancer types, including liver, large intestine, ovary, central nervous system, etc. 

*mRNA structural changes:* sSNVs can alter mRNA structure (Figure 5), stability [84,85,86], and translational speed [87], potentially causing disease [88,89,90]. RNAsnp [38] is a computational tool to predict whether an SNV induces significant mRNA structural changes. Of our set of 2111 *proposed driver* sSNVs, 104 (Appendix A) were predicted by RNAsnp to cause significant mRNA structural changes. These predicted mRNA structure-changing sSNVs are found in multiple cancer types in our set, e.g., breast, skin, urinary tract, and liver.

*Changes to binding of proteins:* RNA Binding Proteins (RBPs) bind to a specific RNA sequence motif or secondary structure to regulate multiple post-transcriptional events, including mRNA splicing, polyadenylation, localization, and degradation [91,92]. To date, over 1500 RBPs have been identified [93], which bind to a motif that is 3–7 nucleotide long [42]. Oncogenic effects of mutations in RBP-coding genes have been well documented [92]. Mutations in cancer-associated RBP binding sites can alter RNA expression and splicing [94]. Notably, Teng et al. experimentally demonstrated that sSNVs can disrupt the binding between RBP and the transcripts of cancer genes (e.g., *DAB2* and *PCBP3*, *ZFHX3* and *PTBP1*) [95]. Here, we extracted all putative RBP motifs in all human protein coding sequences using the FIMO (Find Individual Motif Occurrences) [40] and examined whether these motifs overlap with our *proposed driver* sSNVs. We identified 107 genomic-based *proposed driver* sSNVs that overlap with RBP binding motifs (Appendix A). 

*Changes to transcription factor binding:* Another possible oncogenic effect of cancer driver mutations is alteration of transcription factor binding sites (TFBS) [96,97]. We used the SNP2TFBS tool [43] to find *proposed driver* sSNVs mapping to TFBS. However, none of our variants were labeled as TFBS-affecting. 

Of the 2111 genomic position-based *proposed driver* variants, our functional analysis identified 326 sSNVs of specific impact mechanisms (Figure 6; 136 sSNVs affecting splice sites, 104 sSNVs inducing RNA structural changes, and 107 sSNVs affecting RBP motifs; some variants with multiple impacts). These 326 sSNVs (genomic position-based; 609 transcript-based) are primarily found in skin, large intestine, lung, and liver cancers (Appendix A) in our set. The functional impacts of other *proposed driver* sSNVs require further investigation. Note that our pipeline for putative driver sSNV selection and evaluation of results are inherently limited by the accuracy of the computational tools (synVep, RNAsnp, spliceAI, and CADD-splice, and FIMO) used in the analysis. Additionally, some driver mutations may fail to pass our recurrence filters due to low frequency or high tumor heterogeneity [98]. Finally, it is also possible that the *proposed driver* sSNVs do not individually act as cancer driver mutations, but collectively contribute to cancer progression [8,52,53].

## 4. Conclusions

Here, we developed and evaluated a new way to identify sSNV cancer drivers and proposed a means of tagging cancer genes (for a graphical overview, see Appendix A). To identify drivers, we used variant recurrence in cancer data, together with synVep functional impact scores and variant position conservation. We showed that our *proposed driver* variants selected in this manner are enriched in known cancer genes and pathways. However, they also identify genes that have not previously been deemed relevant to cancer. We further found that a higher number of putative drivers per gene is likely an indication of that gene’s involvement in cancer appearance and/or progression. Finally, we showed that at least 15% of our putative driver variants likely disrupt cellular mechanisms known to be cancer associated. Our results highlight the potential importance of synonymous variants in causing cancer. Our methods may also be used in prioritizing experimental validation of cancer driver sSNVs and novel cancer genes in the future.

## Figures and Tables

**Figure 1 genes-13-00778-f001:**
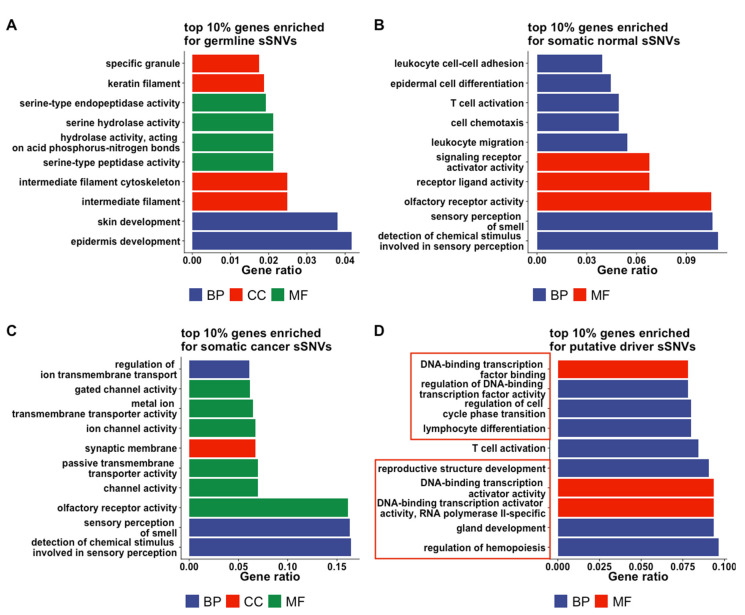
**Genes enriched for different categories of sSNVs differ in GO terms.** (GO) terms from resulting from gene-set enrichment analysis (GSEA) are shown for (**A**) germline, (**B**) somatic normal, (**C**) somatic cancer, and (**D**) putative driver sSNVs. The *X*-axis (gene ratio) is the percentage of the input genes that are associated with the specific GO term. The bars are colored by GO term groups, i.e., BP: biological processes in blue, CC: cellular component in green, MF: molecular function in red. Only the top 10 GO terms are shown for each category of sSNVs. The GO terms that are specific to the putative driver category are shown in red boxes in panel (**D**).

**Figure 2 genes-13-00778-f002:**
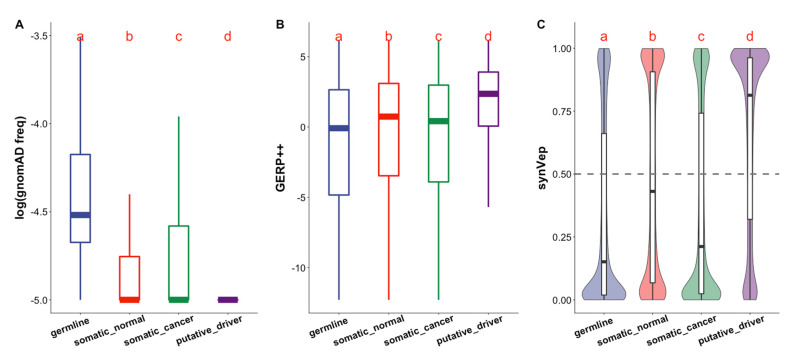
**Variation in population frequency, conservation, and synVep predictions of the *germline*, *somatic normal*, *somatic cancer*, and *putative driver* sSNVs.** Variant types are indicated by color: germline is blue, somatic normal is red, somatic cancer is green, and putative driver is purple. Variants differ by (**A**) population frequency (gnomAD frequencies; sSNVs with 0 frequency are set to log (freq) = −5 for display purposes), (**B**) conservation (GERP++ scores), and (**C**) effects (synVep predictions; scores > 0.5, i.e., above the gray dashed line, indicate effect). For each panel, the Kruskal–Wallis test rejected (*p*-value < 2 × 10^−16^) the null hypothesis that all groups follow the same distribution; as did the Dunn test pairwise comparisons (different letters indicate statistically different distributions).

**Figure 3 genes-13-00778-f003:**
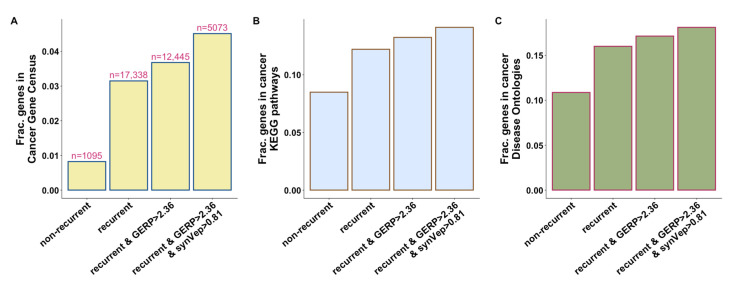
**Genes identified by recurrence, conservation, and synVep prediction are more likely to be involved in cancer.** The four categories of genes are identified by the filters (*X*-axis) as described in text. *Y*-axis represents the fraction of the selected category genes that are found in (**A**) Cancer Gene Census, (**B**) KEGG cancer pathway genes, and (**C**) DO cancer genes. Numbers on top of each bar in panel A show the number genes of each category.

**Figure 4 genes-13-00778-f004:**
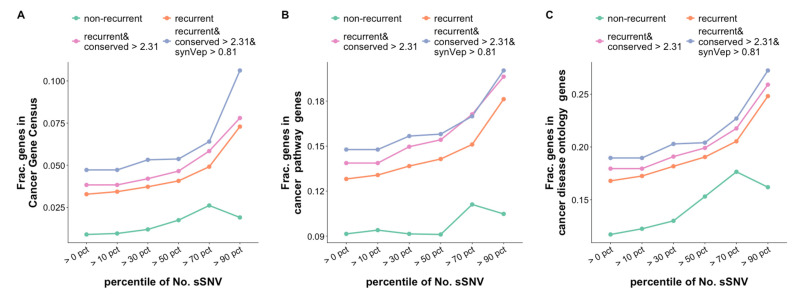
**Genes with more sSNVs are more likely to be involved in cancer.** The four categories of genes are identified by the above-described filters (Methods). The *X*-axis indicates that genes with more than the corresponding amount of sSNVs (represented as percentile) are selected. *Y*-axis represents the fraction of the selected category genes that are found in (**A**) CGC, (**B**) KEGG cancer pathway genes, and (**C**) DO cancer genes. Numbers on top of each bar in panel A show the number genes of each category.

**Figure 5 genes-13-00778-f005:**
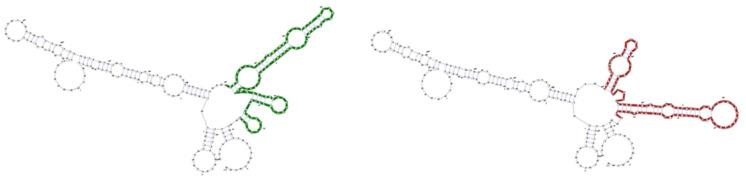
**An example of RNA structural change due to an sSNV.** A transcript (ENST00000371081) experiences structural change from wildtype (green) to mutant (red) due to a proposed driver sSNV (C165A). The illustration is generated by RNAsnp [38] web server (https://rth.dk/resources/rnasnp/ (accessed on 12 March 2022)).

**Figure 6 genes-13-00778-f006:**
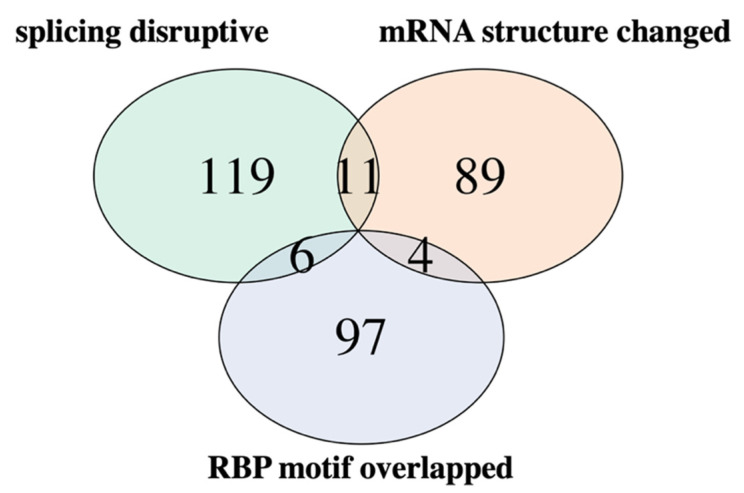
**Venn diagram of the sSNV functional impacts.** Count of sSNVs affecting splicing (predicted by CADD-splice and spliceAI), RBP motifs, or changing mRNA structure. Pairwise hypergeometric tests suggest that the overlaps between variant sets are not statistically significant (*p*-value > 0.05).

## Data Availability

Transcript sequences used for RBP motif extraction can be found in figshare https://figshare.com/articles/dataset/transcript_sequences_zip/19407530 (accessed on 24 March 2022).

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
