# Peer review of "Inferring Potential Cancer Driving Synonymous Variants"

_genes, 2022, doi:10.3390/genes13050778_

Round 1

Reviewer 1 Report

The paper "Inferring potential cancer driving synonymous variants" by Zishuo Zeng and Yana Bromberg describes an original study on the role of synonymous single nucleotide variants (sSNVs) in cancer, and a procedure to recover this often overlooked category of genomic events. It is a very interesting study but my main criticism is that their analysis, conducted with the interesting synVEP tool, could yield much more information if some more digging out is performed. Below, my points:

- The study describes the prevalence of sSNVs in specific pathways (defined in Gene Ontology, KEGG). But it would very, very interesting if they could highlight which genes are more affected by sSNVs, divided in the groups 1) germline, 2) somatic normal, 3) somatic cancer (of course the fourth group  used in the current analysis, "putative driver", would be enriched for cancer drivers, so I suggest merging it with the other groups).

- Related to the previous point, I was really surprised to find only a brief mention of three example genes (PCDH15, CELF4 and MYBPC1), rather than a full analysis ranking the genes most affected by sSNVs in different groups. This would answer extremely relevant questions such as: are common tumor suppressors (e.g. PTEN, TP53) affected by a higher burden of synonymous variants than other genes? Or: are oncogenes more or less affected by synonymous mutations? What are the genes most affected by sSNVs? Of course this number should be normalized appropriately (by gene length, by the frequency of non-synonymous mutations affecting the gene, et cetera).

- Again, related to the previous point, the list shown in Supplementary Table 1 should be expanded by 1) dividing the observed sSNVs by mutation class (germline, somatic normal, somatic cancer), 2) normalizing by gene length (e.g. nr sSNVs/1000nts), 3) showing the frequency of nonsynonymous SNVs as a comparison. Currently, the list is not very informative and it describes only the tip of the analysis. Also, the list would be of very high interest in the main text in my opinion, and not relegated in the Supplementary.

- It seems like a wasted opportunity the choice to not investigate synonymous short indels (those happening in the non-coding regions of transcripts, for instance) with their pipeline. If the authors don't include them in their analysis, they justify this lack of inclusion and specify the limitation of their study.

- Although it is already published, the authors should better describe their tool in the context of the study, synVEP (starting from explaining the acronym), since the bulk of their analysis resides on this algorithm.

- A flow chart, or analogous graphical summary, should be added, indicating the application of synVEP and the structure of the pipeline.

- Line 28: the concept of "Cancer hallmark" requires an explanation, or a citation.

Reviewer 2 Report

In this report by Zeng and Bromberg et al, the authors have developed an interesting tool called synVep, a machine learning to predict the effect of synonymous mutations in cancer. The authors found recurrent sSNV (Synonymous single nucleotide variants) among cancer patients. The authors use the COSMIC database data to analyze a large number of samples for their tool prediction and identify 326 sSNVs which are predicted to have effects on RNA structure, RNA splicing, and mutations in RBP motifs. The authors propose the predicted sSNVs provide computational guidance for experimental validation of synonymous mutations in cancer. This is an interesting tool developed by the authors and addresses a very vital question of passenger mutations' effect on cancer. Even though the manuscript has merits, some concerns if addressed will significantly improve the manuscript.

Did the authors compare with any other tool that is there in the field to show how their tool is better?

The authors have shown the Venn diagram. The significance of the overlap is missing.

Did the authors try to use dbNSFP v4 (even though it is known for non-synonymous variants) to predict the effect of synonymous mutations that affect splicing and then compare it with their tool?

Can the authors differentiate between solid tumors (breast, lung for example) and liquid tumors (leukemia), based on the number of synonymous mutations? Also, predict which tumors are more deleterious?

Do the authors need to mention the limitations of the tool if there are any? And the conclusion is better discussed.

The authors need to correct GEA to Gene-set enrichment analysis.

Round 2

Reviewer 1 Report

I believe the authors did not answer all my points. A graphical flow chart should be added, indicating the application of synVEP and the structure of the pipeline. I believe this would increase the readability of the paper, giving the reader a more intuitive understanding of the work, as well as improving the following citations. I am sure compiling such a flowchart should not take more than a couple hours to be performed, and I am sure the authors should take this opportunity while the pipeline and the nuances of their article are still fresh in their memory.

Author Response

Thanks for the suggestion. We now added the flowchart as Supplementary Figure 3. We put it in the supplementary materials because we think the flowchart is not very information-rich (the procedures are relatively straightforward). But if the reviewer insists, we will transfer it to the main text.